# Preprocedural Viral Load Effects of Oral Antiseptics on SARS-CoV-2 in Patients with COVID-19: A Systematic Review

**DOI:** 10.3390/biomedicines11061694

**Published:** 2023-06-12

**Authors:** Miriam Ting, Alex Dahlkemper, Jeremy J. Schwartz, Manzel Woodfork, Jon B. Suzuki

**Affiliations:** 1Department of Periodontics, University of Pennsylvania, Philadelphia, PA 19104, USA; 2General Practice Residency, Albert Einstein Medical Center, Philadelphia, PA 19141, USA; abdahlkemper@gmail.com (A.D.); schwartzjj1@gmail.com (J.J.S.); manzel.woodfork@jefferson.edu (M.W.); 3Think Dental Learning Institute, Paoli, PA 19301, USA; 4Department of Graduate Periodontics, University of Maryland, Baltimore, MD 21201, USA; jon.suzuki@temple.edu; 5Department of Graduate Prosthodontics, University of Washington, Seattle, WA 98015, USA; 6Department of Graduate Periodontics, Nova Southeastern University, Ft. Lauderdale, FL 33314, USA

**Keywords:** SARS-CoV, COVID, mouthwash, antiseptics, cetylpyridinium, chlorhexidine, iodine, peroxide, antiviral, rinses

## Abstract

(1) There are limited clinical trials to support the effectiveness of mouth rinses when used as a preprocedural rinse against severe acute respiratory syndrome coronavirus 2 (SARS-CoV-2). This systematic review aims to evaluate the efficacy of antiseptic mouth rinses as a preprocedural rinse in reducing SARS-CoV-2 oral viral load *in-vivo*. (2) Methods: A literature search was conducted through November 2022 for the following databases: PubMed, Web of Science, Cochrane Library, and Google Scholar. The evaluated outcomes were quantitative changes in viral load and the statistical significance of that change after using antiseptic mouth rinses. (3) Results: 14 randomized controlled trials (RCT) were selected for risk of bias assessment and data extraction. (4) Conclusion: Within the limits of this systematic review, preprocedural mouth rinses may significantly reduce SARS-CoV-2 in the mouth, thus, reducing the viral particles available for airborne dispersion. Preprocedural mouth rinses may be an effective strategy for reducing airborne SARS-CoV-2 dispersion in the environment. Their use may be a preventive strategy to reduce the spread of COVID-19 in selected medical and healthcare facilities, including dental clinics. Potential preprocedural mouth rinses are identified for use as an integral part of safe practice for healthcare protocols. This systematic review was registered with the National Institute for Health Research, international prospective register of systematic reviews (PROSPERO): CRD42022315177.

## 1. Introduction

The severe acute respiratory syndrome coronavirus 2 (SARS-CoV-2) transmission has been linked to exhaled aerosols [1]. Viral load generating this mode of transmission of SARS-CoV-2 is highest in the oropharynx, nasopharynx, and nasal cavity [2]. Aerosols can be generated during speech, breathing, coughing, and sneezing. Patients that are pre-symptomatic, symptomatic or asymptomatic are all potential sources of transmission [3,4]. Infection control practices have been critical in controlling the infection and the spread of the virus [5]. Vaccination has been a key role in reducing SARS-CoV-2 spread [6]; however, the vaccines have failed to effectively exclude infectious virus in the upper respiratory tract [7]. Oral antiseptics have been reported to reduce the risk of disease transmission and viral infectivity from aerosol-generating procedures. 

Saliva has been used for detecting SARS-CoV-2 in patients with COVID-19, reporting a detection rate of up to 91.7% [8]. Patients with COVID-19 had the highest salivary viral load during the first week after symptom onset when evaluating endotracheal aspirate and saliva samples [9]. This may explain the rapid spread of the pandemic and warrant the use of mouth rinses as a preventive health measure [10].

This suggested that saliva from the oral cavity can be a potential high-risk route of infection for SARS-CoV-2 [11]. The involvement of saliva in SARS-CoV-2 spread suggested that antimicrobial mouthwashes containing substances with virucidal activity can help reduce viral transmission in high-risk environments [12]. This includes household and healthcare settings that perform aerosol-generating procedures. Virucidal effects have been seen in substances contained in oral antiseptics [13,14,15]. Interestingly, animal models observing SARS-CoV-1, a virus similar in genomics to SARS-CoV-2, showed that the virus persisted in oral mucous membranes for up to 2 days before infecting the lower respiratory tract [16]. This may present an opportunity to control disease progression with a potential therapeutic antiseptic rinse applicable to the oral cavity.

Promising data in both *in vitro* and clinical studies have shown the virucidal effects of antimicrobial mouthwashes or oral antiseptics against viruses such as influenza [17]. *In vitro*, contact of 15–60 s with different oral antiseptics resulted in solid virucidal effects and reduced viral infectivity [18,19,20]. Specifically relating to coronaviruses, evidence of virucidal effects against Middle East Respiratory Syndrome Coronavirus (MERS-CoV) and SARS-CoV-1 has been seen with povidone-iodine (PVP), an active ingredient in several oral antiseptics [21,22]. Another substance, cetylpyridinium chloride (CPC), was reported to decrease more than a thousand times the infectivity of SARS-CoV-2 measured by the tissue culture [23]. This activity was confirmed in distinct variants of SARS-CoV- 2, suggesting a broad antiviral efficacy. In addition, chlorhexidine (CHX) and hydrogen peroxide (H_2_O_2_) have *in vitro* virucidal effects against coronaviruses and SARS-CoV-2 [19,20,24,25,26,27].

Antiseptic mouth rinses have been adopted into the pre-appointment protocols of many dental offices due to the COVID-19 pandemic. This is based empirically on the available outcome of *in vitro* studies at the start of the pandemic to reduce the oral viral load and the possibility of transmission [27,28]. However, extremely limited clinical trials were available then to support the effectiveness of the mouth rinses on SARS-CoV-2. This systematic review aims to evaluate which mouth rinses effectively reduce oral SARS-CoV-2 shedding in the oral cavity and assess the substantivity of the mouth rinses to suppress the viral load.

## 2. Materials and Methods

### 2.1. Focus Question

The focus question is, “What are the clinical effects of oral antiseptics on SARS-CoV-2 in patients with COVID-19?”. The PICOS parameters are Participants: patients infected with SARS-CoV-2; Interventions: oral antiseptic mouthwashes; Comparisons: use of saline or water or compared to baseline; Outcomes: antiviral effects; Study Design: clinical studies.

### 2.2. Literature Search and Study Design

PubMed, Web of Science, Cochrane Library, and Google Scholar were searched until November 2022. Google Scholar was used for the grey literature search. The following keywords (Table 1) were used to search each database: “SARS-CoV-2”, “COVID-19”, “coronavirus”, “oral antiseptics”, “mouthwash”, “chlorhexidine”, “iodine”, “peroxide”, “cetylpyridinium”, “alcohol”, “chlorine dioxide”, “povidone”, “octenidine dihydrochloride”, “polyaminopropyl biguanide”, “chloride”. The reference list of the selected articles was further hand-searched for any articles not included in the initial search. Authors of the selected articles were contacted to request additional quantitative data or information regarding their studies. The corresponding authors of the selected studies were given one week to respond with any additional information. 

This systematic review was registered with the National Institute for Health Research, international prospective register of systematic reviews (PROSPERO): CRD42022315177. There were no amendments to the initial PROSPERO registered information. This systematic review was conducted in accordance with the PRISMA (Preferred Reporting Items for Systematic Reviews and Meta-Analyses) guidelines [29]. The flowchart (Figure 1) illustrates the systematic literature search conducted according to the PRISMA guidelines [29]. 

### 2.3. Inclusion Criteria

Randomized controlled trialsClinical studies on SARS-CoV-2 onlyOnly studies reporting quantitative data on viral loadOnly mouthwashes are used in the oral cavity

### 2.4. Exclusion Criteria

*In-vitro* studies were excludedDescriptive studies were excludedCase reports were excludedNasal irrigation/sprays were excludedStudies reporting qualitative data were excludedStudies reporting data as % of SAR-CoV-2 positive patients were excluded

### 2.5. Screening and Data Extraction

The “Title and Abstract” were independently screened by two reviewers (M.T. and M.W.); articles were excluded if they did not meet the inclusion criteria. The full text was independently analyzed by two reviewers (M.T. and M.W.) and verified by another two reviewers (J.J.S. and A.D.). A previously pilot-tested data extraction sheet was used by one reviewer (M.T.) for data extraction and independently verified by two other reviewers (J.J.S. and A.D.).

### 2.6. Risk of Bias Assessment

The risk of bias (Table 2) was assessed using the revised tool for assessing the risk of bias in randomized trials (RoB 2) [30]. Two reviewers (J.J.S. and A.D.) independently scored the risk of bias for the selected studies. Any disagreements were resolved with a discussion with a third reviewer (M.T.). 

## 3. Results

### 3.1. Search Results

The search yielded 8633 studies: 421 in Cochrane Library, 3700 in Google Scholar, 96 in PubMed, 4416 in Web of Science, and no additional articles from hand searching of the reference list of the selected articles (Figure 1). After the title and abstract screening, the duplicates were removed, and 50 articles remained for full-text analysis. After full-text analysis, 36 were eliminated: 6 for not being a randomized trial [44,45,46,47], 3 for being an *in vitro* study [48,49,50], 17 for not having quantitative data of viral load [51,52,53,54,55,56,57,58,59,60,61,62,63,64,65,66,67], 3 for not being in the oral cavity [68,69,70], 5 for not being a mouth rinse [71,72,73,74,75], and 2 for being a summary protocol [76,77]. Finally, 14 articles were selected for data extraction [12,31,32,33,34,35,36,37,38,39,40,41,42,43].

### 3.2. Quality of Evidence

The risk of bias (Table 2) of the selected 14 randomized controlled trials reported mostly low risk of bias. Ten of the 14 studies have a low risk of bias [12,31,32,33,34,35,38,40,41,43], three studies have one domain with unclear risk of bias [36,37,42], and 1 study has two domains with unclear risk of bias or high risk of bias [39]. All selected studies utilized baseline or control or both for comparison. However, selected studies with controls were too heterogenous to enable a meaningful meta-analysis. The quantitative data reported utilized different units of measure, monitored different sampling time points, different rinse protocols, different follow-up periods, different mouth rinses and different combinations of mouth rinses. 

### 3.3. Study Characteristics

All 14 selected studies were randomized controlled trials (Table 3). Many of these studies assessed viral load in saliva by measuring viral RNA by means of reverse transcription-quantitative polymerase chain reaction (RT-qPCR). The included studies conducted randomized controlled trials of patients infected with SARS-CoV-2. The outcomes reported were viral load pre-intervention and viral load at designated time intervals post-intervention.

Mouth rinses evaluated in the selected studies were PVP-I (0.5%, 1%, 2%), H_2_O_2_ (1%, 1.5%), CPC (0.7%, 0.75%, CHX (0.12%, 0.2%, beta-cyclodextrin and citrox (CDCM), CPC + Zn, H_2_O_2_ + CHX, HCIO (0.02%), BAC. The controls used in the studies were either distilled water, water, saline or an unreported placebo. Patients in the selected studies rinsed 10–20 mL of an antiseptic mouth rinse for 30–60 s. Three studies did an additional rinse again some minutes later [33,34,41]. The follow-up duration ranges from immediate to 6 h after rinsing. Six of the 14 studies reported no adverse reaction from the mouth rinses [12,33,34,35,39,42]. The other eight studies did not report adverse reactions [31,32,36,37,38,40,41,43].

### 3.4. Outcomes 

The timeline reporting a significant difference in SARS-CoV-2 reduction compared to the baseline in the selected studies is illustrated in Figure 2. Timepoints unmarked in Figure 2 were either not evaluated or did not report significant changes.

CPC, CHX, H_2_O_2_ and distilled water showed a significant reduction of SARS-CoV-2 up to 60 min compared to baseline. PVP-I and saline showed a significant reduction of SARS-CoV-2 up to 45 min compared to baseline. CDCM showed a significant reduction of SARS-CoV-2 up to 540 min compared to baseline. H_2_O_2_ + CHX and CPC + Zn only reported a significant reduction of SARS-CoV-2 immediately post-rinse.

Thus, CPC, CHX, H_2_O_2_, PVP-I, CDCM, saline and distilled water can significantly affect viral reduction up to 45–60 min compared to baseline. However, further evaluation beyond 60 min is needed to report on the long-term efficacy.

Saline and water appear to have a significant effect in the reduction of SARS-CoV-2 compared to baseline. It is important to distinguish if mouth rinses are significantly better than water and saline alone. 

The timeline reporting a significant difference in SARS-CoV-2 reduction compared to control in the selected studies is illustrated in Figure 3. Timepoints unmarked in Figure 3 were either not evaluated or did not report significant changes. 

CPC showed a significant reduction of SARS-CoV-2 at 5 min, 30 min, 180 min and 360 min compared to controls (distilled water and water). PVP-I showed a significant reduction of SARS-CoV-2 at 5 min, 30 min, 45 min and 360 min compared to controls (distilled water and saline). CHX showed a significant reduction of SARS-CoV-2 at 5 min and 60 min compared to controls (placebo and saline). HCIO showed a significant reduction of SARS-CoV-2 at 30 min compared to the control (saline). CDCM showed a significant reduction of SARS-CoV-2 at 240 min compared to the control (placebo).

Our analyses demonstrated that PVP-I and CPC are effective for up to 360 min, CDCM for up to 240 min, CHX for up to 60 min, and HCIO for up to 30 min. Data for significant viral reduction for CHX beyond 60 min and HCIO beyond 30 min has not been reported.

## 4. Discussion

Preprocedural antiseptic rinses are suggested for COVID-19 management and prevention. Public health officials generally agree that one of the primary vehicles of SARS-CoV-2 transmission and infection is human aerosol particles containing SARS-CoV-2 particles. 

These guidelines were encouraged to be followed, especially by medical and dental offices seeing potential COVID-19 patients. In addition to these guidelines, strategies to minimize the risk of COVID-19 transmission from patient to healthcare provider need to be explored. Potentially cost-effective and efficient methods to reduce oral SARS-CoV-2 are preprocedural mouth rinses for all patients seen by medical (e.g., ENT, anesthesiologists, audiologists, speech therapists) and dental healthcare providers.

This paper identifies antimicrobial mouth rinses which have been reported to have significant effects on the reduction of airborne SARS-CoV-2 particles.

### 4.1. Effect and Substantivity of Oral Antiseptics

Mouth rinses have historically been used to reduce cross-contamination from aerosolized bacteria [78]. They have been reported effective against herpes, influenza, parainfluenza, and hepatitis B [55]. This suggests mouth rinses could significantly mitigate SAR-CoV-2 in the oral cavity [55].

The use of mouth rinses to remove SARS-CoV-2 in the oral cavity does not equate to treatment for COVID-19, nor does it provide a permanent reduction in oral viral load. SARS-CoV-2 can replicate in the respiratory system, and virus-infected secretions from the upper respiratory tract can reinfect the oral cavity during speech or coughing. 

Most mouth rinses reduce SARS-CoV-2 viral load at 15 min post-rinse, with the persistence of viral load reduction extending to 45 min [33]. Mouth rinses reduce the viral load by disrupting the viral envelope and by mechanical flushing of the virus. The SARS-CoV-2 envelope consists of lipids and glycosylated proteins, which can be disrupted by cationic and amphiphilic compounds in mouth rinses [17]. The spike glycoprotein of SAR-CoV-2 is inserted in the viral envelope. Disruption of the viral envelope causes increased permeability and neutralization of SARS-CoV-2 [79]. Antiseptic mouth rinses physicochemically disrupt and destroy the SARS-CoV-2 viral lipid envelope [80]. Experimental evidence using density gradient ultracentrifugation followed by RT-qPCR showed that antiviral activity against coronaviruses is mostly caused by disruption of the viral envelope [81]. Thus, mouth rinses may be a safe and effective option to reduce SARS-CoV-2 transmission from the mouth.

However, the exposure time of SARS-CoV-2 to the mouth rinses *in vivo* depends on the salivary clearance and the substantivity of the mouth rinse. The antiviral effects of the mouth rinses can be diluted by the patient’s salivary flow *in vivo* [82].

PVP-I was proposed as a preoperative gargle in COVID-19 cases to mitigate the spread of SARS-CoV-2 among healthcare workers [83]. PVP-I application was reported to suppress SARS-CoV-2 RNA in saliva in the short-term [33,84]. It was reported to inactivate up to 99.99% of SARS-CoV-2 in 30 s [35]. Other concentrations of PVP-I (0.5%, 1%, and 1.5%) also reported similar inactivation of SARS-CoV-2 even at 15 s [85]. This effect could extend up to 3 h [36,86]. However, PVP-I did not sustain a prolonged oral and nasal SARS-CoV-2 RNA reduction 8 h after application [68]. It is reported that PVP-I can be used safely for up to 6 months in the mouth [36,85,86,87,88,89].

The half-life of CPC in the mouth ranged from 3–5 h after rinsing [90]. This finding appears consistent with prior investigations showing a potential persistence of activity between 180 and 300 min [91]. CPC can adhere to the oral mucosa for several hours. However, CPC failed to show substantivity and duration of activity to the level of CHX [92]. 

CPC showed antiviral activity and the potential to inactivate the COVID-19 coronavirus intraorally [93]. It continues to be active against viral particles increasing the viral nucleocapsid levels between 1–3 h. CPC can have synergistic effects and increase substantivity with other active ingredients. CPC has less side effects than rinses containing CHX, such as tooth staining, taste disruption, burning sensation, and mouth ulcers though mostly aesthetic effects [94,95].

CPC combined with O-cymen-5-ol extended the substantivity by more than 1 h and significantly improved antibacterial effects compared to CPC alone [96]. Because CPC shows synergistic potential on substantivity with other active ingredients, it may be a focus of future studies to assess the effects of CPC and CHX in combination and the resultant substantivity on SARS-CoV-2.

Antimicrobial effects associated with CPC and CHX are linked to the cationic state of the substance that allows binding to negatively charged oral proteins and surface proteins of antigens [97,98]. However, it should be noted that the inactivation of these cationic substances can be brought about by anionic compounds such as the detergents used in oral hygiene products.

CHX-containing mouth rinses showed short-term reductions in SARS-CoV-2, which returned to baseline levels in 2 h [99]. Significant reductions of viral load by CHX were reported at timelines up to 60 min [35]. Thus, CHX is substantive for at least 60 min [34]. Conflicting reports suggest that CHX weakly inactivates coronaviruses [81]. And may indicate that saliva interactions and mouth rinse concentration may influence the efficacy of the mouth rinse. A reduction of salivary pH has been reported to decrease the effects of CHX [100]. Reduced pH has electrostatic drug-protein effects that decrease the protein binding ability of CHX [101]. The concentration of the CHX solution may affect the effectiveness pharmacologically. Dental plaque can act as a reservoir for CHX and prolong its effects in biofilm [102]. The ability of saliva to function as a reservoir for CHX can be a factor for antimicrobial effects [103,104]. The concentration of CHX may influence the magnitude of such effects. It has been demonstrated that a 0.2% concentration of CHX has more antimicrobial activity than 0.12% concentrations [105,106].

Moreover, CHX shows greater potential for reservoirs at higher concentrations. There was a significant difference in plaque inhibition of 0.2% CHX concentration compared to 0.12% concentration [107]. Based on CHX’s ability with respect to concentration as a reservoir for plaque, one may theorize that CHX concentration may have comparative effects as a reservoir in salvia and produce antimicrobial effects. CHX lozenges showed some beneficial effects, and the higher CHX concentration as it dissolves in saliva may produce greater antiviral activity [108].

H_2_O_2_ disrupts the viral envelope and degrades the viral RNA via oxidative properties [109]. H_2_O_2_ effectively reduces SARS-CoV-2 titers in saliva immediately after rinsing [35]. However, the viral load returned to baseline values within 60 min after rinsing [35]. Thus, the lack of substantivity is a disadvantage of H_2_O_2_ as a mouth rinse. 

In adults with asymptomatic or mild COVID-19, CDCM had a significant effect on SARS-CoV-2 salivary viral load reduction 4 h after the initial dose [32]. However, the data on the long-term benefits of CDCM and its effects on patients with high salivary load is limited.

Using a combination of 2 or more antiseptic agents may improve the antiviral capacity. However, antagonistic effects leading to decreased antiviral impact cannot be excluded. COVID-19 patients rinsing with H_2_O_2_ followed by CHX resulted in a minimal reduction in salivary viral load; the compaction was minimal even immediately post-rinse [35]. The secondary CHX rinse may have washed out the H_2_O_2_ before there was sufficient contact time for antiviral activity. It is possible the outcomes can be improved by rinsing with CHX first, and then H_2_O_2_ since CHX has greater substantivity than H_2_O_2_. There are potential benefits to the sequential use of different types of mouth rinses. However, further research is needed to evaluate synergistic mouth rinse combinations that produce maximal antiviral effects.

### 4.2. Clinical Outcomes Compared to In Vitro

The reported *in vitro* virucidal effects of mouth rinses against various coronaviruses resulted in mouth rinses being recommended during the pandemic to curb the transmission of SARS-CoV-2 [81,110]. Based on *in vitro* studies, PVP-I 0.5–1.5% and CPC were more effective mouth rinses in reducing viral load and inactivating COVID-19 than CHX and H_2_O_2_ [93]. 

*In vivo*, PVP-I was effective against SARS-CoV-2 at the lowest concentration of 0.5% PVP-I and at the lowest contact time of 15 s. These effects were like those seen *in vitro*, where similar concentrations (PVP-I of 0.5%, 1.25%, and 1.5%) were seen to inactivate SARS-CoV-2 at both 15 and 30 s. Other *in vitro* studies have shown a concentration of PVP-I as low as 0.23% was effective at inactivating SARS-CoV-2 in 15 s. 

CPC at a concentration as low as 0.5% *in vitro* can completely inactivate the SARS-CoV-2 virus in as little as 15 s. *In vivo*, CPC was shown to reduce viral load, remain in saliva, and retain its effectiveness due to its high substantivity [93]. CPC has been effective both in the presence and absence of saliva [111].

CHX-containing mouth rinses and H_2_O_2_ demonstrated lower virucidal properties than CPC and PVP-I [112]. CHX concentration ranging from 0.12–2% was effective *in vivo* and *in vitro* against SARS-CoV-2. However, conflicting reports showed CHX and H_2_O_2_ independently had no virucidal effects against SARS-CoV-2. 

Furthermore, H_2_O_2_ was less effective as a preprocedural mouth rinse due to its inability to remain active in saliva for an effective period. At concentrations of 1.5% and 3.0% H_2_O_2_ had minimal *in vitro* antiviral effects after 15 or 30 s. Thus, 0.5–1.5% PVP-I may be preferred over H_2_O_2_ for SARS-CoV-2 inactivation during the pandemic [85].

However, *in vivo*, results reported in the selected studies were modest compared to the *in vitro* results [23,113,114,115,116,117,118]. In addition, *in vitro*, results were not consistently reproduced *in vivo*. CHX did not significantly reduce SARS-CoV-2 viral load *in vitro* [25] but showed some effectiveness *in vivo*. Thus, oral rinses should be used in tandem with strict preventive measures.

Some factors contribute to the lack of consistency between *in vitro* and *in vivo* data. Rinsing *in vivo* generates a shearing effect leading to the rupture of viruses and viral cell junctions. This mechanical effect inherent in rinsing is absent *in vitro* studies. Furthermore, saliva *in vivo* contains multiple proteins and glycoproteins, which may modify the effect of mouth rinses. Mouth rinses like H_2_O_2_ can be inactivated by catalase enzymes in saliva. Saliva also contains bacteria which may bind to the active ingredients in the mouth rinses, limiting the concentration needed for virucidal effects. 

Salivary flow can also dilute mouth rinses *in vivo* [119]. The salivary flow rate of 5 mL/min can dilute mouthwash concentration. The intraoral substantivity of mouthwashes may explain differences reported between *in vivo* results compared to *in vitro*. Furthermore, *in vitro* studies use standardized virucidal efficacy tests that do not represent *in vivo* oral cavity antiviral effects.

*In vitro*, studies are mostly conducted at room temperature. This may affect the mouth rinses’ viral activity and virucidal properties compared to body temperature. The oral cavity has a higher mean temperature of 36.6 °C. SARS-CoV-2 may be more stable at room temperature than at the elevated temperature of the oral cavity, which may contribute to some errors in the outcome. 

Many *in vitro* studies utilize distilled water as a control. Distilled water *in vitro* did not produce a virucidal effect [120]. Distilled water may reduce the viral load by reducing viral lipid attachments, but it does not affect viral protein stability or viability.

*In vivo*, studies utilizing distilled water reported a low but significant decrease in viral load [120]. Mechanical forces during rinsing may release viral particles more effectively and affect viral viability by osmotic pressures.

### 4.3. Limitations

#### 4.3.1. Patient Selection and Sample Size

Limitations to the selected studies include small cohort sizes, unreported baseline oral conditions, and a limited range of patients (e.g., only patients with high viral load or limiting to inpatients or severe COVID-19 and asymptomatic patients were not delineated for inclusion). Most of the studies were conducted on symptomatic hospitalized patients. Asymptomatic patients may not be included. Therefore, outcomes may not represent the whole spectrum of COVID-19 patients. The small cohort sizes may be due to the drastic decreases in COVID-19 cases as the community recovers from the infection. In addition, inter-patient variability can also contribute to data variations.

The limited sample size should be expanded to a larger study population for future studies to increase the statistical power. The baseline data should also include periodontal status, oral hygiene, plaque index, gingival index and quantification of salivary flow. In addition, both positive and negative controls should be included for comparison.

#### 4.3.2. Sampling Methods

Sampling methods for the diagnosis of COVID-19 include saliva samples, throat gargle samples, nasopharyngeal swabs, and oropharyngeal swabs. The selected studies utilized the saliva sampling method (self-collected or investigator supervised) to evaluate the effectiveness of the oral rinses.

Saliva sampling methods have the advantage of avoiding invasive procedures, decreasing the risk of nosocomial transmission, screening large populations quickly, and providing an alternative in situations where nasopharyngeal swabs are contraindicated [8]. Saliva samples are self-collected and reliable [121,122,123]. However, patient training, including supervised collection of the first samples, may be needed to avoid interpatient variability. Since unsupervised self-collection is challenging to standardize; therefore, there may be some discrepancies due to the loss of samples and lower viral loads reported. Studies that utilize saliva samples collected by trained investigators may reduce patient bias and variability [37].

Some limitations of the saliva collection technique relate to the time delay in which saliva samples were collected to the time of symptom onset in asymptomatic patients. It is estimated that 30% of patients do not develop symptoms [124,125]. And SARS-CoV-2 infectivity peaks at or before symptom onset [126]. The delay in saliva collection after symptoms onset may present with lower viral salivary concentration to evaluate further reduction affected by the mouth rinses. Salivary viral load decreased, corresponding to the number of days from infection. In substantivity studies of mouth rinses, it may be difficult to confirm if the viral load decrease is due to the mouth rinse or due to the progression of the infection. In previous studies comparing saliva samples and nasopharyngeal samples, when nasopharyngeal samples still reported a low positive for SARS-CoV-2, a negative was reported for the virus in saliva [116].

Furthermore, other methods, like the throat gargle method, may be better than saliva sampling. The saline mouth and throat gargle method is reported to be easily tolerated by patients and a more sensitive test than saliva collection [127]. Throat gargle samples also showed higher viral load values than nasopharyngeal and oropharyngeal swabs [128,129]. Further studies required comparing the SARS-CoV-2 viral load between saliva collection and other sampling methods collected simultaneously in each patient to better assess the use of saliva as a diagnostic tool for SARS-CoV-2 viral load.

#### 4.3.3. Diagnostics

Assessing the viral load alone may not be the most efficient method for assessing the effectiveness of mouth rinses, especially mouth rinses that primarily target the viral envelope and not the viral RNA. Viral RNA may persist in saliva after the disruption of the virus particle due to the protective effects of protein complexes [130]. Viral RNA, embedded in intact viral particles or released from disrupted ones, can be recovered by RNA extraction methods. Thus, viruses detected via real-time PCR (RT-PCR) do not correlate with the presence of complete viral particles [55]. The RT-PCR detection of the SARS-CoV-2 viral load includes both live and non-infective viral particles and does not assess the viability of the virus. Therefore, even though the data may report that mouth rinses have no significant effect on SARS-CoV-2 viral load as measured by quantitative PCR, there is a possibility that the viral particles may be non-viable. RT-PCR technique can only detect RNA copies and not the infectivity of the detected virus fragments. Therefore, the reported viral load did not correspond to the infectious virus.

To assess for SARS-CoV-2 infectivity would require viral isolation assays and cell culture. Without viral cultures, we cannot exclude the possibility that mouth rinses suppressed viable SARS-CoV-2 and not viral RNA. Viral cell culture is the gold standard for assessing viral infectivity. However, viral cell culture cannot only be performed when there is insufficient sample volume or low viral load. Virus recovery is usually possible from samples with high viral loads greater than 106 copies/mL [131,132]. Samples with lower viral loads or limited volume may limit the viability of the culture.

Monkey Vero-E6 lines are widely used to assess SARS-CoV-2 infectivity in viral culture [113,114,133,134]. Since new protocols utilizing human lung cells are being developed [135]; therefore, future studies utilizing human cell lines would be a more realistic infection model. However, culturing SARS-CoV-2 in a cell culture to evaluate active virus replication requires special laboratory conditions such as biosafety level 4. Due to this limitation in culturing SARS-CoV-2, many studies still utilized viral RNA load as a reliable surrogate marker [136].

The increase in nucleocapsid protein levels indicates an increase in the disruption of viral particles with no infectious capacity. However, the possibility of saliva’s natural disruption of the viral particles cannot be excluded [12]. Furthermore, the high variability in the nucleocapsid protein levels may limit the applicability of the results reported.

Mouth rinses alter viral proteins and membranes [113] and may also be cytotoxic to cultured cells used in infection studies. This can limit the evaluation of antiseptics to biocompatible concentrations. Other limitations of the included studies were the lack of analysis on the infectivity of viruses left in the mouth after rinsing and the viruses bound to soft tissue [35].

### 4.4. Biocide Resistance

Most selected studies reported no adverse reaction from the mouth rinses evaluated. The rest of the selected studies did not report on adverse reactions, and any adverse reactions arising from mouth rinse use was unknown. Long-term use of mouth rinses may pose some risks. Minor side effects of antiseptic mouth rinses may include stained teeth, transient loss of taste, or black hairy tongue. Other risks of using mouth rinses are the development of biocide resistance. Mouth rinses containing alcohol should be avoided by patients with alcohol addictions, reformed alcoholics, alcohol allergies, and religious or cultural convictions against alcohol.

The prolonged use of a sublethal dose of mouth rinses may increase the risk of gram-negative bacterial overgrowth and biocide resistance [137,138,139,140,141]. Gram-positive bacteria have a lower minimum inhibitory concentration (MIC) than gram-negative bacteria; the bacterial cell wall of gram-positive bacteria has a higher affinity to antiseptic mouth rinses. Development of biocide resistance was reported for prolonged low-level CHX exposure [142]. Possible mechanisms of biocide resistance include efflux pump dysfunction and mutation of the cell membrane [142].

Biocide resistance has been linked to bacterial cross-resistance to antibiotics; drug resistance was reported after frequent CPC use [143]. Biocide resistance was mostly reported for non-oral bacteria. There were limited studies reported for oral bacteria. Short-term use of mouth rinses did not report non-native bacteria or gram-negative bacterial overgrowth [144]. Long-term use of mouth rinses may increase the risk of biocide resistance, antibiotic cross-resistance, and phenotypic adaptation.

## 5. Conclusions

In addition to the implementation of improved personal protection and engineering enhanced air filtration, preprocedural mouth rinses may be an effective strategy and cost-reduction solution for reducing airborne SARS-CoV-2 dispersion in the environment and be an integral part of safe practice for healthcare protocols. Within the limits of this systematic review, preprocedural rinses may reduce SARS-CoV-2 particles in the mouth of COVID-19 patients, thus, reducing the number of viral particles available for airborne dispersion. Furthermore, the viral neutralization properties and the mechanisms of action of the mouth rinses were poorly understood. Future research on laboratory analysis assays for capsid disassembly and viral uncoating of SARS-CoV-2 exposed to these mouth rinses is needed.

Antiseptic mouth rinses may be a preventive strategy to reduce the spread of COVID-19 in selected medical and healthcare facilities, including dental clinics. Further studies are needed to investigate the clinical effectiveness of mouth rinses in reducing SARS-CoV-2 transmission in the household, medical facilities, and dental clinics.

This paper identifies potential antimicrobial mouth rinses which have significant effects on the reduction of SARS-CoV-2 particles in the oral cavity.

## Figures and Tables

**Figure 1 biomedicines-11-01694-f001:**
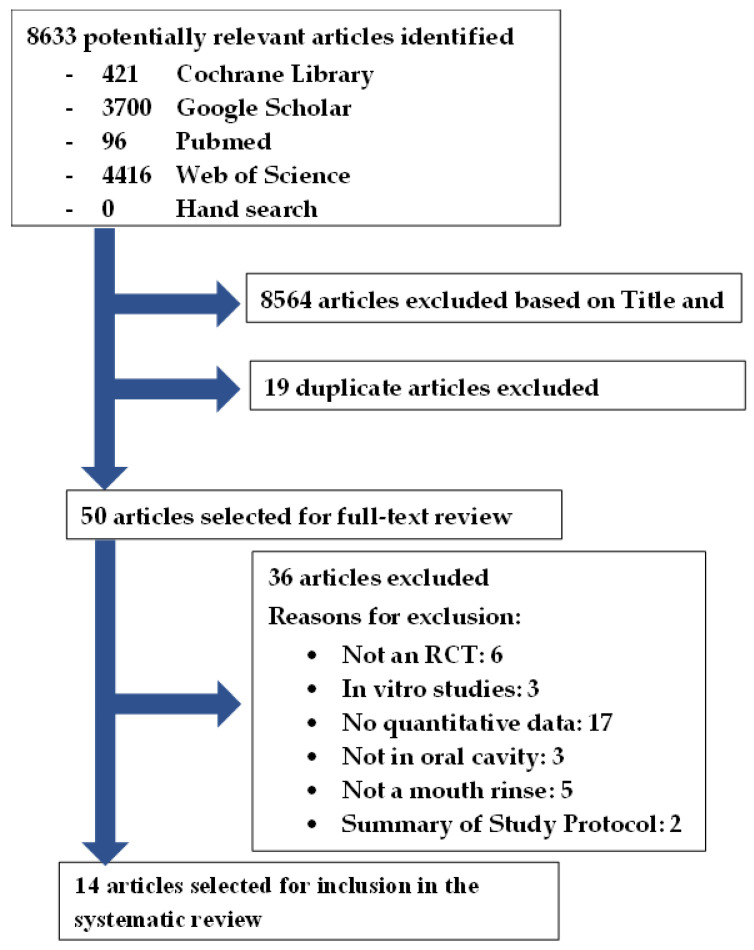
Article selection flowchart in accordance with the PRISMA guidelines [26].

**Figure 2 biomedicines-11-01694-f002:**
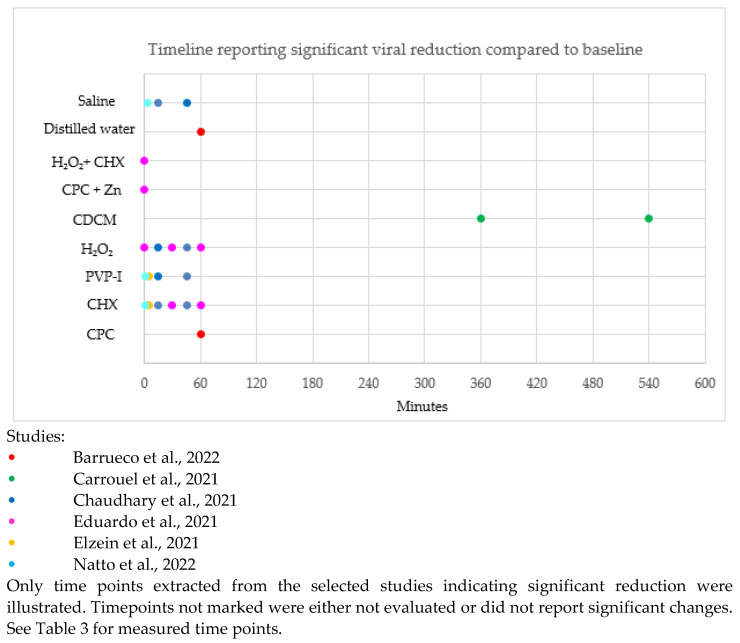
Timeline of significant difference in SARS-CoV-2 reduction after rinsing with antiseptic compared to baseline [31,32,33,35,36,41].

**Figure 3 biomedicines-11-01694-f003:**
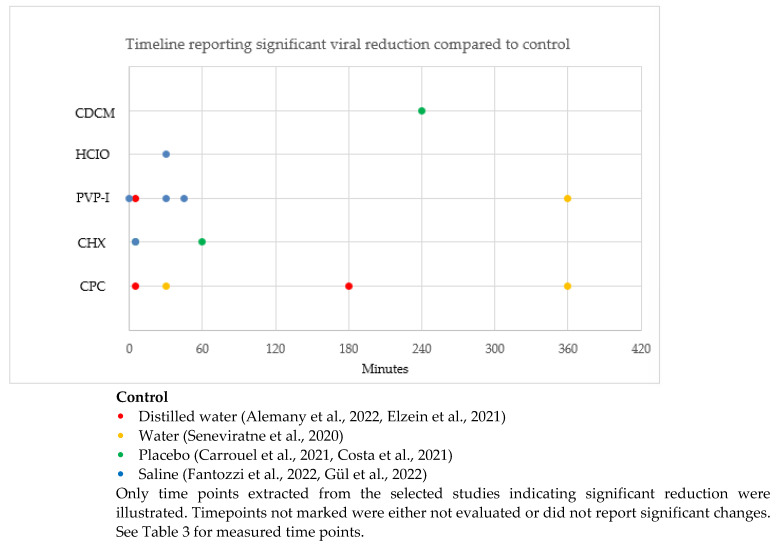
Timeline of significant difference in SAR-CoV-2 reduction after rinsing with antiseptic compared to control [12,32,34,36,37,39,43].

**Table 1 biomedicines-11-01694-t001:** Keyword Search.

Database	Keywords
Cochrane Library	((SARS-CoV-2) OR (COVID-19) OR (coronavirus)) AND ((oral antiseptics) OR (mouthwash) OR (chlorhexidine) OR (iodine) OR (peroxide) OR (cetylpyridinium) OR (alcohol) OR (chlorine dioxide) OR (povidone) OR (octenidine dihydrochloride) OR (polyaminopropyl biguanide) OR (chloride)) in All Text—(Word variations have been searched)—Trials
Google Scholar	with all of the words: mouthwashwith the exact phrase: COVIDwith at least one of the words: antiseptics mouthwash chlorhexidine iodine peroxide cetylpyridinium alcohol chlorine dioxide povidone octenidine dihydrochloride polyaminopropyl biguanide chloridewithout the words: reviewReturn articles dated between 2019–2022
PubMed	((SARS-CoV-2) OR (COVID-19) OR (coronavirus)) AND ((oral antiseptics) OR (mouthwash) OR (chlorhexidine) OR (iodine) OR (peroxide) OR (cetylpyridinium) OR (alcohol) OR (chlorine dioxide) OR (povidone) OR (octenidine dihydrochloride) OR (polyaminopropyl biguanide) OR (chloride))
Web of Science	((SARS-CoV-2) OR (COVID-19) OR (coronavirus)) AND ((oral antiseptics) OR (mouthwash) OR (chlorhexidine) OR (iodine) OR (peroxide) OR (cetylpyridinium) OR (alcohol) OR (chlorine dioxide) OR (povidone) OR (octenidine dihydrochloride) OR (polyaminopropyl biguanide) OR (chloride))

**Table 2 biomedicines-11-01694-t002:** Risk of Bias Assessment 2 (ROB2).

Study	D1	D2	D3	D4	D5
Alemany et al., 2022 [12]	**+**	**+**	**+**	**+**	**+**
Barrueco et al., 2022 [31]	**+**	**+**	**+**	**+**	**+**
Carrouel et al., 2021 [32]	**+**	**+**	**+**	**+**	**+**
Chaudhary et al., 2021 [33]	**+**	**+**	**+**	**+**	**+**
Costa et al., 2021 [34]	**+**	**+**	**+**	**+**	**+**
Eduardo et al., 2021 [35]	**+**	**+**	**+**	**+**	**+**
Elzein et al., 2021 [36]	**-**	**+**	**+**	**+**	**+**
Fantozzi et al., 2022 [37]	**+**	**+**	**+**	**-**	**+**
Ferrer et al., 2021 [38]	**+**	**+**	**+**	**+**	**+**
Gul et al., 2022 [39]	**+**	**-**	**+**	**x**	**+**
Meister et al., 2022 [40]	**+**	**+**	**+**	**+**	**+**
Natto et al., 2022 [41]	**+**	**+**	**+**	**+**	**+**
Redmond et al., 2022 [42]	**+**	**-**	**+**	**+**	**+**
Seneviratne et al., 2020 [43]	**+**	**+**	**+**	**+**	**+**
**Risk of bias domain** D1: Risk of bias arising from the randomization process D2: Risk of bias due to deviations from the intended interventions D3: Risk of bias due to missing outcome data D4: Risk of bias in the measurement of the outcome D5: Risk of bias in the selection of the reported result					+ Low
					- Unclear
					x High

**Table 3 biomedicines-11-01694-t003:** Selected studies.

Study	Aim	No. of Patients	Rinse Protocol	Control	Duration after Rinse	Significance Compared to Control	Significance Compared to Baseline	Adverse Events
Alemany et al., 2022 [12]	To assess the short-term effects of CPC mouthwash on viral RNA load and nucleocapsid protein levels in the saliva of individuals infected with COVID-19	118 Dropout: 11 CPC: 60 Control: 58	Gargle 15 mL for 1 min Active Ingredient: CPC: 0.7%	Distilled water	CPC: Mean viral load Baseline 1 h 3 h Mean Nucleocapsid Baseline 1 h 3 h	NS NS NS NS S (increase) S (increase)	NR	Reported no adverse events
Barrueco et al., 2022 [31]	To test efficacy for already-developed antiseptic formulations	75 Dropout: 31 PVP-I: 9 H_2_O_2_: 6 CPC: 10 CHX: 9 Control: 10	Rinse 15 mL for 1 min Active Ingredient: PVP-I: 2% H_2_O_2_: 1% CPC: 0.7% CHX: 0.12%	Distilled water	Viral Load (RT-qPCR) PVP-I: Baseline 30 min 1 h H_2_O_2_: Baseline 30 min 1 h CPC: Baseline 30 min 1 h CHX: Baseline 30 min 1 h Distilled water: Baseline 30 min 1 h Virus infectivity (Culture) PVP-I: Baseline 30 min 1 h H_2_O_2_: Baseline 30 min 1 h CPC: Baseline 30 min 1 h CHX: Baseline 30 min 1 h Distilled water: Baseline 30 min 1 h	NR	- NS NS - NS NS - NS NS - NS NS - NS S (reduced) - NS (reduced) NS - NS NS - NS S (reduced) - NS NS - NS NS	NR
Carrouel et al., 2021 [32]	To describe the evolution of salivary SARS-CoV-2 viral load in COVID-19 outpatients receiving mouthwashes with or without antivirals	176 beta-cyclodextrin and citrox (CDCM) CDCM: 88 Placebo:88	Rinse 30 mL for 1 min, 3 times/day for 7 days Active Ingredient: beta-cyclodextrin (0.1%) and citrox (0.01%)	Placebo	CDCM: Baseline 4 h 9 h 7 days Placebo: Baseline 4 h 9 h 7 days	- S (reduced) NS NS - - - -	- S (reduced) S (reduced) S (reduced) - S (reduced) S (reduced) S (reduced)	NR
Chaudhary et al., 2021 [33]	To examine the the risk posed by potential patients who report no symptoms of COVID-19 and to investigate the efficacy of a simple intervention (that is, preprocedural mouth rinsing) on reducing salivary viral load	40 Randomly assigned to each group No. in each group, NR	Rinse 15 mL for 30 s followed by another rinse of 15 mL for 30 s Active Ingredient: PVP-I: 2% H_2_O_2_: 1% CHX: 0.12%	Saline	PVP-I Baseline 15 min 45 min H_2_O_2_: Baseline 15 min 45 min CHX: Baseline 15 min 45 min Saline: Baseline 15 min 45 min	- NS (reduced) NS (reduced) - NS (reduced) NS (reduced) - NS (reduced) NS (reduced) - - -	- S (reduced) S (reduced) - S (reduced) S (reduced) - S (reduced) S (reduced) - S (reduced) S (reduced)	Reported no adverse events
Costa et al., 2021 [34]	To evaluate the impact of oral rinsing and gargling with 0.12% chlorhexidine gluconate on the salivary viral load of people infected with SARS-CoV-2	110 Drop out: 10 CHX: 50 Placebo: 50	Gargle 15 mL for 30 s then another rinse of 15 mL for 30 s Active Ingredient: CHX: 0.12%	Placebo(Inactive substance with the same flavor)	CHX: Baseline 5 min 60 min	- S (reduced) S (reduced)	NR	Reported no adverse events
Eduardo et al., 2021 [35]	To determine if commercial products containing 1.5% H_2_O_2_, 0.12% CHX, and 0.075% CPC + 0.28% zinc can reduce the SARS-CoV-2 viral load in the saliva, as well as the time required for oral viral load recovery	60 Dropout: 26 Placebo: 9 H_2_O_2_: 7 CHX: 8 CPC + Zn: 7 H_2_O_2_ + CHX: 12	Placebo: rinse 20 mL for 1 min CPC + Zn: rinse 20 mL for 30 s H_2_O_2_: rinse 10 mL for 1 min CHX: rinse with 15 mL for 30 s H_2_O_2_ + CHX: rinse 10 mL H_2_O_2_ for 1 min, then 15 mL of CHX for 30 s Active Ingredient: H_2_O_2_: 1.5%, CHX: 0.12% CPC + Zn CPC:0.075% Zinc: 0.28% H_2_O_2_ + CHX H_2_O_2_: 1.5%, CHX: 0.12%	Distilled water	H_2_O_2_: Baseline Immediate 30 min 60 min CHX: Baseline Immediate 30 min 60 min CPC + Zn: Baseline Immediate 30 min 60 min H_2_O_2_ + CHX: Baseline Immediate 30 min 60 min Distilled water: Baseline Immediate 30 min 60 min	NR	- S (reduced) S (reduced) S (reduced) - NS S (reduced) S (reduced) - S (reduced) NS NS - S (reduced) NS NS - NS NS NS	Reported no adverse events
Elzein et al., 2021 [36]	To evaluatethe virucidal efficacy of 2 preprocedural mouth rinses: 0.2% Chlorhexidine and 1% Povidone-iodine in the reduction of salivary SARS-CoV-2 viral load	77 Dropout: 16 CHX: 25 PVP-I: 27 Control: 9	Gargle or rinse for 30 s Active Ingredient: CHX: 0.2% PVP-I: 1%	Distilled water	CHX: Baseline 5 min PVP-I: Baseline 5 min Distilled water: Baseline 5 min	- S (reduced) - S (reduced) - -	- S (reduced) - S (reduced) - NS	NR
Fantozzi et al., 2022 [37]	To assess the effectiveness of three different oral antiseptics (chlorhexidine 0.12%, povidone-iodine 1%, hydrogen peroxide 1%) in reducing the oral and oropharyngeal SARS-CoV-2 viral loads	38 PVP-I: 8 H_2_O_2_: 11 CHX: 8 Control: 11	Rinse and gargle 15 mL for 60 s Active Ingredient: PVP-I: 2% H_2_O_2_: 1% CHminX: 0.12%	Saline (NaCl 0.9%)	Viral load PVP-I Baseline Immediate 45 min H_2_O_2_: Baseline Immediate 45 min CHX: Baseline Immediate 45 min Saline: Baseline Immediate 45 min	NS S (reduced) S (reduced) NS NS NS NS NS NS - - -	- NS - - NS - - NS - - NS -	NR
Ferrer et al., 2021 [38]	To test whether any of these standard oral antiseptics appear to diminish viral load in saliva and could be used to reduce transmission risk in clinical and social settings.	84 Dropout: PVP-I: 9 H_2_O_2_: 14 CPC: 11 CHX: 12 Control: 12	Rinse for 1 min Active Ingredient: PVP-I: 2% H_2_O_2_: 1% CPC: 0.7% CHX: 0.12%	Distilled water	PVP-I: Baseline 30 min 60 min 120 min H_2_O_2_: Baseline 30 min 60 min 120 min CPC: Baseline 30 min 60 min 120 min CHX: Baseline 30 min 60 min 120 min Distilled water: Baseline 30 min 60 min 120 min	- NS NS NS - NS NS NS - NS NS NS - NS NS NS - - - -	- NS NS NS - NS NS NS - NS NS NS - NS NS NS - NS NS NS	NR
Gül et al., 2022 [39]	To clinically evaluate the effect of the hypochlorous acid (HClO) and PVP-I solutions on the viral load of SARS-CoV-2 in COVID-19 patients	75 Dropout:14 HCIO: 20 PVP-I: 21 Control: 20	Gargle 20 mL for 30 s Active Ingredient: HCIO: 0.02% PVP-I: 0.5%	Saline	PVP-I Baseline 30 min HCIO Baseline 30 min Saline Baseline 30 min	- S (reduced) - S (reduced) - -	- NS - NS - NS	Reported no adverse reaction
Meister et al., 2022 [40]	To investigate the most effective compound in a randomized placebo-controlled clinical trial for its efficacy in terms of reducing viral loads and infectivity in the oral the cavity of infected individuals	24 BAC: 24 Control: 6	Gargle 15 mL Active Ingredient: BAC	Saline	Baseline 15 min 30 min	NR	NR	NR
Natto et al., 2022 [41]	To assess the short-term efficacy of over-the-counter mouth rinses and lozenges in minimizing the salivary viral load of SARS-CoV-2 in COVID-19 patients when compared with saline	45 PVP-I: 15 CHX: 15 Control: 15	Rinse 10 mL for 30 s, and then again 5 min later Active Ingredient: PVP-I: NR CHX: NR	Saline	PVP-I Baseline After rinse CHX Baseline After rinse Saline Baseline After rinse	- NS - NS - -	- S (reduced) - S (reduced) - S (reduced)	NR
Redmond et al., 2022 [42]	To evaluate the efficacy of oral and nasal povidone-iodine in reducing the burden of SARS-CoV-2 RNA in patients with COVID-19	22 Dropout: 4 PVP-I: 10 Control: 8	Gargle Active Ingredient: PVP-I: 1%	Phosphate-buffered saline	PVP-I: Baseline 8 h 24 h Saline Baseline 8 h 24 h	- NS NS - - -	- NS NS - NS NS	No adverse effects reported
Seneviratne et al., 2020 [43]	To evaluate the efficacy of three commercially available mouth rinses, namely PI, CHX and CPC, on the salivary SARS-CoV-2 viral load in a cohort of SARS-CoV-2-positive patients in Singapore	16 PVP-I: 4 CPC: 4 CHX: 6 Control: 2	Rinse for 30 s PVP-I (10 mL) CHX (15 mL) CPC (20 mL) Control (20 mL) Active Ingredient: PVP-I: 0.5% CPC: 0.75% CHX: 0.2%	Water	PVP-I: Baseline 5 min 3 h 6 h CPC: Baseline 5 min 3 h 6 h CHX: Baseline 5 min 3 h 6 h Water: Baseline 5 min 3 h 6 h	- NS NS S (reduced) - S (reduced) NS S (reduced) - NS NS NS - - - -	- NS NS NS - NS NS NS - NS NS NS - NS NS NS	NR

NR: Not reported, NS: Not significant, S: Significantly different.

## Data Availability

Not applicable.

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
