# Peer review of "Preprocedural Viral Load Effects of Oral Antiseptics on SARS-CoV-2 in Patients with COVID-19: A Systematic Review"

_biomedicines, 2023, doi:10.3390/biomedicines11061694_

Round 1
Reviewer 1 Report
This is a well-written, well-researched, scholarly, carefully crafted review of preprocedural mouthwashes for prevention of COVID. Such mouthwashes in commonly used during the heights of the pandemic. A better understanding of the risks and benefits of these mouthwashes could guide practice recommendations for future control of viral transmission in procedural settings. The scientific interest and potential audience for this paper is reasonably large. I have a few addressable concerns, which I summarize below.
MAJOR
Need to make clear that this is a review of the ability of mouthwashes to reduce viral load and is not a review of the ability to mouthwashes to have a benefit on transmission or health outcomes of COVID. To that end, the phrase "Clinical effects" should not be in the title of the manuscript. Perhaps the new title should be something like
"Preprocedural viral load effects of oral antiseptics on SARS-CoV-2 in patients with COVID-19: A systematic review"
Other than a single sentence about CPC, side effects are not mentioned in this article. There needs to be at least a brief discussion on the risks/benefits of mouthwashes. Some (or all) of these have toxicity. SO if the authors are suggesting a benefit from their use (reduction in SARs-CoV-2) they also need to highlight that any such benefit could come at some risk. Then do their best to quantify that risk, compare the risks, and or state that the risk are unknown or not possible to evaluate.
Comparisons to other reviews on the same topic needs to be made. Why is the review different? Why should the reader want to read this review in addition to (or instead) of the other review(s). A PubMed search for
"oral antiseptic" AND COVID AND review
reveals at least one other review that should be cited in the Introduction as part of the above discussion.
And this PubMed search
mouthwash AND "SARS-COV-2" AND review
reveals a large number of manuscripts, at least some of which seem like they should be part of this Introductory paragraph as well
MINOR
Add 'dental' to Title
Add 'dental' to Abstract
or perhaps 'preprocedural' rather than 'dental'
Otherwise some readers will think that this review is relevant to everyday life, such as "Should I use mouthwash before going grocery shopping?". See also the first Major Concern, above.
"As an alternative, oral antiseptics have been reported to reduce the risk of disease transmission, and the viral infectivity from aerosol generating procedures."
Change to (because these are in addition to, not as an alternative):
"Oral antiseptics have been reported to reduce the risk of disease transmission, and the viral infectivity from aerosol generating procedures."
Table 3 "doesn't work". That is, it is hard to read. Too much white space; too long. I think you need to remove a lot of the information from the table to make it easier to read. Put the removed information into the prose body of the manuscript. And/or move some information to a Supplemental Table. Table length should be at most one printed page.
Figure 2 & 3. Add a symbols showing negative timepoints in addition to the positive timepoints. Could be something like gray Xs to distinguish them from the positives. It is hard to interpret the positive results without negative result context.
"Our data demonstrated..."
rephrase. It is not your data. Maybe something like
"Our analyses demonstrated..."
"Preprocedural antiseptic rinses are important for COVID-19 management and prevention."
I don't see how your metaanalysis has demonstrated this.
You have shown that rinses can kill some virus in the mouth, but have not data or analyses that show this inhibits transmission or clinical outcomes.
"At the inception of the COVID-19 outbreak in early 2020, government officials from the CDC, NIH, and the FDA recommended 6 feet distancing between individuals and face masks to decrease risk of COVID-19 aerosol transmission."
I think you can delete this sentence. It is well known. It is does not seem relevant to this manuscript.
"These guidelines were encouraged to be followed, especially by medical and dental offices seeing potential COVID-19 patients. Therefore, strategies to minimize the risk for COVID-19 transmission from patient to healthcare provider need to be explored."
How does the second sentence follow from the first? Is there some sort of logic that justifies the use of the word "therefore"?
"One of the most cost effective and efficient methods to reduce risk are preprocedural mouth rinses for all patients seen by medical (e.g., ENT, anesthesiologists, audiologists, speech therapists) and dental healthcare providers."
Reduce risk of what? Infection? Clinical outcomes? All you have shown is viral titer reduction. You have not shown any risk reduction.
"This paper identifies antimicrobial mouth rinses which have been reported to have significant effects on reduction of airborne SARS-CoV-2 particles."
I agree. This is the key. This is your bread and butter. Focus on this conclusion in your Discussion.
"Viral Cell culture is the gold standard..."
->
"Viral cell culture is the gold standard..."
"In addition to implementation of improved personal protection and engineering enhanced air filtration."
Sentence fragment.
"Within the limits of this systematic review, pre-procedural rinses (according to the manufacturer’s directions) can be an effective strategy and cost-reduction solution for reducing airborne SARS-CoV-2 dispersion in the environment and be an integral part of safe practice for healthcare protocols."
Sort of strange to include the parenthetical phrase here "(according to the manufacturer’s directions)" considering that manufacturer's directions were not a major aspect of the paper before. I think it is a given that most drugs should be used "as intended". I would delete this phrase.
"Laboratory analysis assays for capsid disassembly and viral uncoating of SARS-CoV-2 exposed to these mouth rinses with longer follow-up to evaluate substantivity can add to the knowledge."
seems a bit like word salad.
what does it mean to "evaluate substantivity"
"can add to the knowledge" seems a bit vague. Everything can add to knowledge, so maybe just delete this clause
OR think about what you really are trying to convey and rewrite the whole sentence so that all readers get your point
Author Response
MAJOR
Need to make clear that this is a review of the ability of mouthwashes to reduce viral load and is not a review of the ability to mouthwashes to have a benefit on transmission or health outcomes of COVID. To that end, the phrase "Clinical effects" should not be in the title of the manuscript. Perhaps the new title should be something like
"Preprocedural viral load effects of oral antiseptics on SARS-CoV-2 in patients with COVID-19: A systematic review"
Thank you for the important insights, the title is amended as suggested.
Other than a single sentence about CPC, side effects are not mentioned in this article. There needs to be at least a brief discussion on the risks/benefits of mouthwashes. Some (or all) of these have toxicity. SO if the authors are suggesting a benefit from their use (reduction in SARs-CoV-2) they also need to highlight that any such benefit could come at some risk. Then do their best to quantify that risk, compare the risks, and or state that the risk are unknown or not possible to evaluate.
Done
Comparisons to other reviews on the same topic needs to be made. Why is the review different? Why should the reader want to read this review in addition to (or instead) of the other review(s). A PubMed search for
This is different because this review is a systematic review. The other reviews are narrative reviews. Reviews on this topic before 2021, may not contain sufficient clinical data as COVID-19 was first identified in Dec 2019. And at least 1 year is needed for IRB approvals and conduct of clinical studies.
"oral antiseptic" AND COVID AND review
reveals at least one other review that should be cited in the Introduction as part of the above discussion.
And this PubMed search
mouthwash AND "SARS-COV-2" AND review
reveals a large number of manuscripts, at least some of which seem like they should be part of this Introductory paragraph as well
Some more recent reviews were included in the Introduction section and cited as suggested. The following reviews were cited:
Guerrero Bernal CG, Reyes Uribe E, Salazar Flores J, Varela Hernández JJ, Gómez-Sandoval JR, Martínez Salazar SY, Gutiérrez Maldonado AF, Aguilar Martínez J, Lomelí Martínez SM. Oral Antiseptics against SARS-CoV-2: A Literature Review. Int J Environ Res Public Health. 2022 Jul 19;19(14):8768. doi: 10.3390/ijerph19148768. PMID: 35886619; PMCID: PMC9316971.
Chumpitaz-Cerrate V, Chávez-Rimache L, Ruiz-Ramirez E, Franco-Quino C, Erazo-Paredes C. Evaluation of Current Evidence on the Use of Oral Antiseptics Against SARS-CoV-2: A Narrative Review. J Int Soc Prev Community Dent. 2022 Oct 31;12(5):488-499. doi: 10.4103/jispcd.JISPCD_65_22. PMID: 36532329; PMCID: PMC9753926.
Weber J, Bonn EL, Auer DL, Kirschneck C, Buchalla W, Scholz KJ, Cieplik F. Preprocedural mouthwashes for infection control in dentistry-an update. Clin Oral Investig. 2023 Apr 20:1–12. doi: 10.1007/s00784-023-04953-z. Epub ahead of print. PMID: 37079156; PMCID: PMC10116478.
MINOR
Add 'dental' to Title
Add 'dental' to Abstract
or perhaps 'preprocedural' rather than 'dental'
Done
"As an alternative, oral antiseptics have been reported to reduce the risk of disease transmission, and the viral infectivity from aerosol generating procedures."
Change to (because these are in addition to, not as an alternative):
"Oral antiseptics have been reported to reduce the risk of disease transmission, and the viral infectivity from aerosol generating procedures."
Done
Table 3 "doesn't work". That is, it is hard to read. Too much white space; too long. I think you need to remove a lot of the information from the table to make it easier to read. Put the removed information into the prose body of the manuscript. And/or move some information to a Supplemental Table. Table length should be at most one printed page.
Table 3 is improved
Figure 2 & 3. Add a symbols showing negative timepoints in addition to the positive timepoints. Could be something like gray Xs to distinguish them from the positives. It is hard to interpret the positive results without negative result context.
Figure 2 &3 are timepoints where the selected studies reported statistically significant reduction in viral assay. Viral reduction data from selected studies were either significantly reduced or not significant. So there were no negative timepoints.
"Our data demonstrated..."
rephrase. It is not your data. Maybe something like
"Our analyses demonstrated..."
Amended as suggested
"Preprocedural antiseptic rinses are important for COVID-19 management and prevention."
You have shown that rinses can kill some virus in the mouth, but have not data or analyses that show this inhibits transmission or clinical outcomes.
Amended to “Preprocedural antiseptic rinses are suggested for COVID-19 management and prevention.”
"At the inception of the COVID-19 outbreak in early 2020, government officials from the CDC, NIH, and the FDA recommended 6 feet distancing between individuals and face masks to decrease risk of COVID-19 aerosol transmission."
I think you can delete this sentence. It is well known. It is does not seem relevant to this manuscript.
Done
"These guidelines were encouraged to be followed, especially by medical and dental offices seeing potential COVID-19 patients. Therefore, strategies to minimize the risk for COVID-19 transmission from patient to healthcare provider need to be explored."
How does the second sentence follow from the first? Is there some sort of logic that justifies the use of the word "therefore"?
Sentence amended and “therefore” removed
"One of the most cost effective and efficient methods to reduce risk are preprocedural mouth rinses for all patients seen by medical (e.g., ENT, anesthesiologists, audiologists, speech therapists) and dental healthcare providers."
Reduce risk of what? Infection? Clinical outcomes? All you have shown is viral titer reduction. You have not shown any risk reduction.
Amended to “Potentially cost effective and efficient methods to reduce oral SARS-CoV-2 are preprocedural mouth rinses for all patients seen by medical (e.g., ENT, anesthesiologists, audiologists, speech therapists) and dental healthcare providers.”
"This paper identifies antimicrobial mouth rinses which have been reported to have significant effects on reduction of airborne SARthis S-CoV-2 particles."
Focus on this conclusion in your Discussion.
Thank you. Amendments made as suggested
"Viral Cell culture is the gold standard..."
->
"Viral cell culture is the gold standard..."
Done
"In addition to implementation of improved personal protection and engineering enhanced air filtration."
Sentence fragment.
Sentence improved.
"Within the limits of this systematic review, pre-procedural rinses (according to the manufacturer’s directions) can be an effective strategy and cost-reduction solution for reducing airborne SARS-CoV-2 dispersion in the environment and be an integral part of safe practice for healthcare protocols."
Sort of strange to include the parenthetical phrase here "(according to the manufacturer’s directions)" considering that manufacturer's directions were not a major aspect of the paper before. I think it is a given that most drugs should be used "as intended". I would delete this phrase.
Done
"Laboratory analysis assays for capsid disassembly and viral uncoating of SARS-CoV-2 exposed to these mouth rinses with longer follow-up to evaluate substantivity can add to the knowledge."
"can add to the knowledge" seems a bit vague. Everything can add to knowledge, so maybe just delete this clause
OR think about what you really are trying to convey and rewrite the whole sentence so that all readers get your point
Done

Reviewer 2 Report
The authors made an attempt to assess the effectiveness of common mouth rinses in reducing SARS-Co-2 viral load in patients diagnosed with COVID-19 using a systematic review approach. The study is well designed, primary consent seems scientifically sound and well evidence-based, however, some issues should be raised.
Major comments.
The electronic databases should include EMBASE as well
Was there any reason not to perform meta-analysis which could significantly improve a scientific value of the study?
Did all authors fully disclosed the conflict of interest considering the assets from companies producing mouthwashes? Did authors ever been involved in lectures promoting such agents?
Aliitaton
Five self-citations for authors consideration
Minor comments
Overoptimistic statement regarding the usefulness of commonly use mouth rinses in prevention of COVID-19 needs more realistic alteration and adjustment considering clinical settings and highly ununiform protocol of reviewed studies .
Rephrasing in some paragraphs contained in Discussion
Author Response
The electronic databases should include EMBASE as well
Thank you for the important comments.
At least 2 major databases are required for a systematic review. We have included 4 databases. We would have used EMBASE if we had access to it. We did not have a subscription for the paid database EMBASE.
Was there any reason not to perform meta-analysis which could significantly improve a scientific value of the study?
The selected studies with controls were too heterogenous to enable a meaningful meta-analysis. The quantitative data reported, utilized different units of measure, monitored different sampling timepoints, different rinse protocols, different follow-up periods, different mouth rinses and different combinations of mouth rinses.
Did all authors fully disclosed the conflict of interest considering the assets from companies producing mouthwashes? Did authors ever been involved in lectures promoting such agents?
Conflict of interest disclosed.
One author may have included mouth rinses in some lectures but only in a general sense for disease mitigation. That author did not receive honoraria for endorsement of any antimicrobial rinses.
Affiliations are listed in manuscript submission.
Overoptimistic statement regarding the usefulness of commonly use mouth rinses in prevention of COVID-19 needs more realistic alteration and adjustment considering clinical settings and highly ununiform protocol of reviewed studies . Rephrasing in some paragraphs contained in Discussion
Alterations and rephrasing made.

Reviewer 3 Report
Why there is this comment in the submitted article? Commented [MT1]: Harrel SK and Molinari JA. Aerosols and splatter in dentistry A brief review of the literature and infection control implications. J Am Dent Assoc. 2004; 135:430-437
was it submitted also to another journal?
The article is well structured.
It respects all the suggested indications for writing a scientific article.
Why there is this comment in the submitted article? Commented [MT1]: Harrel SK and Molinari JA. Aerosols and splatter in dentistry A brief review of the literature and infection control implications. J Am Dent Assoc. 2004; 135:430-437
was it submitted also to another journal?
The article is well structured.
It respects all the suggested indications for writing a scientific article.
Author Response
Why there is this comment in the submitted article?
Thank you for your insightful comments and kind remarks.
The comment in the submitted article was an error and has been removed.
was it submitted also to another journal?
The manuscript was never submitted to another journal.

Round 2
Reviewer 1 Report
Figures 2 and 3 need to be re-imagined.
Consider Chaudhary et al. (2021) for example. These authors ONLY looked at two timepoints: 15 minutes and 45 minutes. So we don't know what happens after 50 minutes (much less 600 minutes). Your graphs make it seem like there is some reason to suspect that there is no effect of H2O2 after an hour. But we just don't know that because it was not measured. At the least, you need to make some indication on this graph of when the last measured timepoint was for each of these compounds. Maybe the answer is a some sort of multiple bar chart, or a table. As it stands, these graphs are misleading.
Author Response
Thank you for the very important observations.
We agree that Figure 2 and 3 needed clarification as to the marked timepoints.
The following sentences were added to the Figure 2 and 3 to further clarify the figures:
"Only timepoints extracted from the selected studies indicating significant reduction were illustrated. Timepoints not marked were either not evaluated or did not report significant changes. See Table 3 for measured timepoints."
Similar clarification was also added to the manuscript, please see tracked changes
